# Feasibility of Using SWIR-Transformed Reflectance (STR) in Place of Surface Temperature (Ts) for the Mapping of Irrigated Landcover

Mohammad Abuzar [1,*], Kathryn Sheffield [1] and Andy McAllister [2]

1   Agriculture Victoria Research, Department of Energy, Environment and Climate Action (DEECA), AgriBio, 5 Ring Road, Bundoora, VIC 3083, Australia; kathryn.sheffield@agriculture.vic.gov.au

2   Agriculture Victoria Research, Department of Energy, Environment and Climate Action (DEECA), 255 Ferguson Road, Tatura, VIC 3616, Australia; andy.mcallister@agriculture.vic.gov.au

*   Correspondence: mohammad.abuzar@agriculture.vic.gov.au

**Abstract:** (1) Background: A simple approach to map irrigated landcover has been introduced by using measures derived from the optical spectral range as an alternative to the thermal range. It has been demonstrated that substituting surface temperature (Ts, 'thermal approach') with SWIR-transformed reflectance (STR, 'optical approach') to detect surface moisture is feasible with comparable results. (2) Methods: Using an iterative thresholding procedure to minimize within-class variance, the bilevel segmentation of variables derived from Landsat-8 representing surface moisture and vegetation cover was achieved for the 2020–2021 summer for a key irrigation district in Australia. (3) Results: The results of irrigated landcover by the optical approach were found to be comparable with those obtained by the thermal approach. The classification accuracy was assessed using water delivery records at the farm level. Although the overall accuracy was high in both cases, the optical approach (97.6%) performed slightly better than the thermal approach (93.9%). (4) Conclusions: The feasibility of using STR to map irrigated landcover has been confirmed by a high-level overall accuracy assessment. This has broader implications in terms of irrigated landcover assessment, as the use of satellite imagery in these applications may not necessarily be limited to microwave or thermal sensors.

**Keywords:** irrigated landcover; thresholding method; Central Goulburn District

## 1. Introduction

Irrigation is critical for agricultural production, and it plays an important role in food security in many areas of the world [1–3]. Irrigation not only increases the crop yield by overcoming climatic constraints, but it also modifies local climatic conditions and adds to evapotranspiration [4,5]. In addition, irrigation can have some environmental impacts like increased soil salinity [6], changes in the water table [7] and altered hydrological cycles [8]. The stated economic and ecological importance of irrigation warrants that the extent and distribution of irrigation is mapped precisely and monitored regularly.

Soil moisture is a key indicator of surface irrigation and irrigated landcover. The interaction of electromagnetic (EM) radiation with soil moisture and irrigated land has shown a significant correlation at various wavelengths. As a result, several methods have been developed to map surface irrigation in different EM regions, namely, optical, thermal and microwave. These methods have been summarized in detail in several recent studies [9–12]. As microwave sensors are particularly sensitive to the changes in ground moisture, these have been used in mapping irrigated landcovers in many studies [13]. Though microwave sensors have provided promising results for the detection of irrigation in many areas, the coarse spatial resolution and the confounding effects of topography and vegetation created notable uncertainties in some regions [14]. However, the launch of the Sentinel-1 satellites constellation (S1-A and S1-B) provided an unprecedented radar data

coverage at a high spatial and temporal resolution. This has led to some new approaches for mapping irrigation based on machine learning and empirical modelling, providing improved results [15,16].

Thermal infrared (TIR) sensors are considered highly relevant for irrigation mapping [17,18]. To distinguish between 'irrigated' and 'non-irrigated' land, the TIR-derived surface temperature (Ts) has been widely used [19,20]. Ts is invariably used in combination with certain vegetation measures to map irrigated agriculture. There are very many vegetation indices which have been developed since the early attempts in the 1960s–1970s [21,22]. Vegetation indices frequently use a combination of spectral responses between visible and the near- or mid-infrared range [23]. The most used vegetation measure is the normalized difference vegetation index (NDVI), which is based on the near-infrared (NIR) and red wavebands. Presently, there are many satellites with optical sensors onboard, which provide images suitable for vegetation indices at medium to high spatial resolution. However, there is a lack of satellites that provide thermal images with acceptable resolution, except for the Landsat series of satellites (presently Landsat-8/9), which continues to provide concurrent thermal and optical observations at a nominal resolution of 30 m.

Although the thermal-based methods are powerful for irrigation mapping since they have physical rationale for their application, most methods depend on local weather conditions such as air temperature (Ta) close to the time of the satellite overpass. The relationship between the Ts and soil moisture is highly influenced by the Ta and other atmospheric conditions [24]. Without the calibration of the Ts, the results are likely to vary spatially across different landcovers, confounding the distinction of irrigated landcover. Considering this limitation, efforts have been made to find a robust substitute of Ts in the optical range to adequately account for surface moisture. Several indices that utilize optical observations have been proposed for the quantification of surface moisture or the lack of it, as shown in Table 1. Some of these indices are based on triangular or trapezoidal pixel distributions of optical observations in different spectral frequency ranges. It is notable that these optical indices (Table 1) are mostly empirical and lack any physical foundation. However, in a recent development, a method which is supported by both theory and experimental data has been proposed by Sadeghi et al. [25]. This new method, which has been designed to measure soil moisture, is based on a linear model in shortwave infrared (SWIR) wavelengths, termed as 'SWIR-transformed reflectance' (STR). We chose this measure to use in this study to explore its feasibility as a substitute of the TIR-derived surface temperature for the mapping of irrigated land cover in the Central Goulburn Irrigation District (CGID) of Victoria, Australia, during the peak irrigation season (i.e., the summer) of 2020–2021.

**Table 1.** Soil moisture and drought indices based on reflectance in optical range.

| Index | Equation * | Reference |
|---|---|---|
| Vegetation Condition Index, VCI | $VCI = 100 * \frac{(NDVI - NDVI_{min})}{(NDVI_{max} - NDVI_{min})}$ | Kogan, 1995 [26] |
| Normalized Difference Water Index, NDWI | $NDWI = \frac{R_{Green} - R_{NIR}}{R_{Green} + R_{NIR}}$ | McFeeters, 1996 [27] |
| Normalized Multiband Drought Index, NMDI | $NMDI = \frac{R_{860\ nm} - (R_{1640\ nm} - R_{2130\ nm})}{R_{860\ nm} + (R_{1640\ nm} - R_{2130\ nm})}$ | Wang and Qu, 2007 [28] |
| Perpendicular Drought Index, PDI | $PDI = \frac{1}{\sqrt{M^2+1}}(R_{Red} + MR_{NIR})$ | Ghulam et al., 2007a [29] |
| Modified Perpendicular Drought Index, MPDI | $MPDI = \frac{R_{Red} + MR_{NIR} - fv(R_{v,Red} + MR_{v,NIR})}{(1-fv)\sqrt{M^2+1}}$ | Ghulam et al., 2007b, [30] |
| Modified Shortwave Infrared Perpendicular Water Stress index, MSPSI | $MSPSI = \frac{1}{\sqrt{1+M^2}}(R_s + MR_d)$ | Feng et al., 2013 [31] |
| Distance Drought Index, DDI | $DDI = \frac{\sqrt{R_{Red}^2 + R_{NIR}^2}}{1+NDVI}$ | Yang et al., 2008 [32] |
| Visible and Shortwave Infrared Drought Index, VSDI | $VSDI = 1 - (R_{SWIR} + R_{Red} - 2R_{Blue})$ | Zhang et al., 2013 [33] |
| Shortwave Infrared Water Stress Index, SIWSI | $SIWSI = \frac{R_{SWIR} - R_{NIR}}{R_{SWIR} + R_{NIR}}$ | Fensholt & Sandholt, 2003 [34] |

* *NDVI*, *NDVI_{min}*, *NDVI_{max}*: smoothed weekly normalized difference vegetation index, its multi-year absolute maximum and minimum, respectively; *R*: reflectance; *R_{Green}* and *R_{NIR}*: reflectance of green and near-infrared spectral bands; *M*: slope of the soil line on *R_{Red}* and *R_{NIR}* space; *fv*: estimated vegetation cover; *Rv*: coefficient taken as 0.5; and *R_{SWIR}*: reflectance of shortwave infrared spectral band.

The objectives of this study were (1) to assess the feasibility of using STR for mapping irrigated landcover and (2) to compare the TIR-based landcover classification (thermal approach) with the STR-based landcover (optical approach).

*Study Area*

The CGID is located approximately between 36°04′41″ S and 36°38′26″ S latitude, and between 144°46′37″ E and 145°23′41″ E longitude in the northern part of Victoria, Australia (Figure 1). It is spread across the council jurisdiction of the Greater Shepparton City. It covers an area of approximately 1,900 sq. km., of which a large part is irrigated. This area is an alluvial plain, dominated by fluviatile sedimentation since the Early Tertiary period. The sediment deposits vary from approximately 50 m to 125 m in depth. The well-drained soils are red-brown, mainly fine-sandy loams. The poorly drained soils are grey heavy soils.

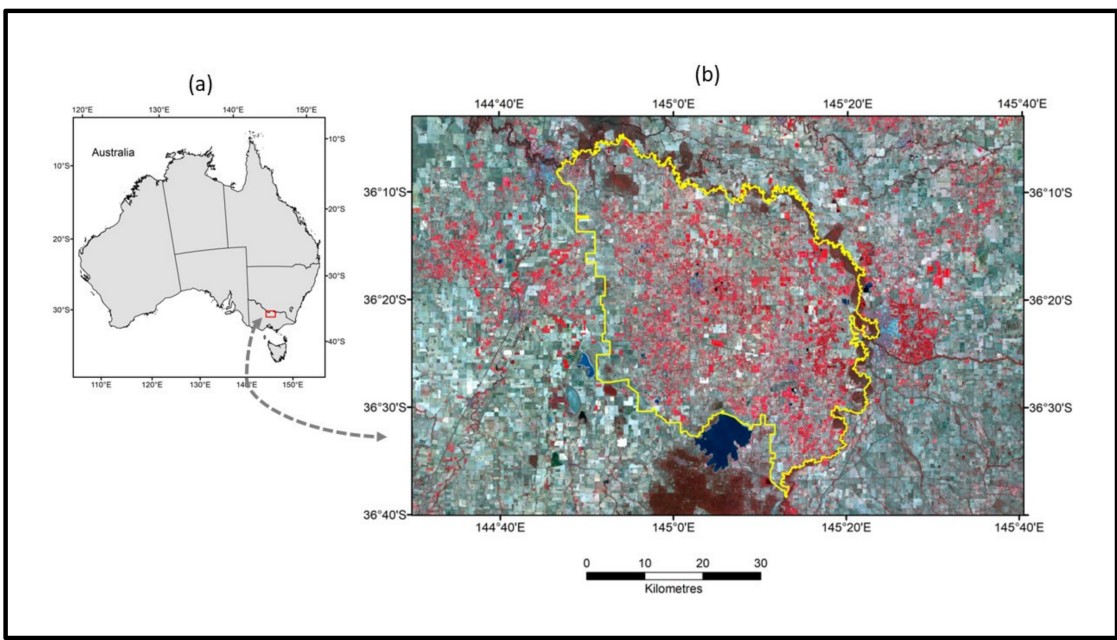

**Figure 1.** The study area. (**a**) CGID location in Australia, and (**b**) extent of CGID (in yellow) with the background of Landsat-8 satellite image bands 5/4/3 (displayed as a false colour composite image).

The climate is temperate, and the region is relatively dry with an average annual rainfall between 200 mm and 400 mm. In general, most of the rainfall is received during winter (June to August). Summers (December to February) are usually dry. The average maximum temperature ranges between 12° and 15 °C in winter, increasing to between 27° and 30 °C in summer (December to February). The average minimum temperature varies from 3° to 6 °C in winter and from 12° to 15 °C in summer (Figure 2). Victoria's largest river, the Goulburn, flows on the eastern boundary before it joins Australia's largest river, the Murray, in the north. The flat terrain is covered by a network of irrigation channels. The irrigation systems used in the region include micro-irrigation, conventional sprinklers, flooding and furrowing. The main industries are dairying, horticulture, cropping and grazing.

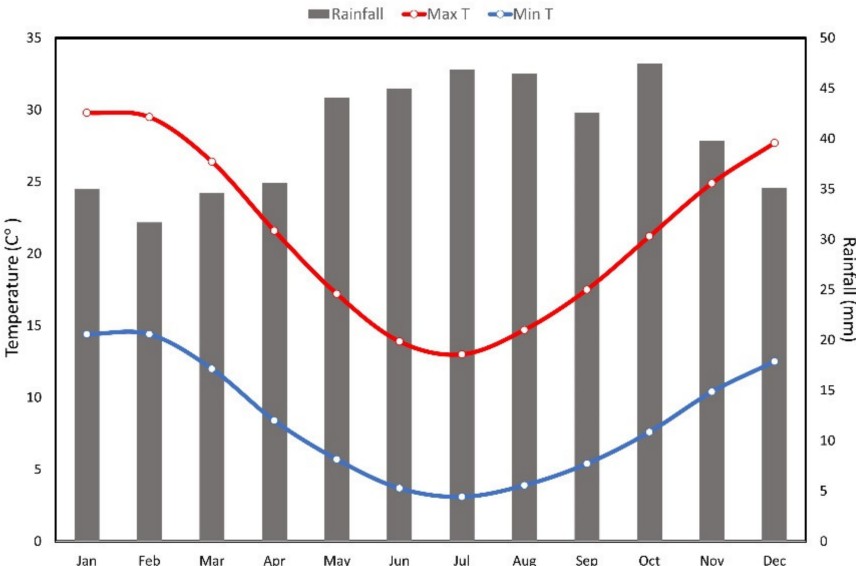

**Figure 2.** Average monthly rainfall and minimum/maximum temperatures at the Tatura weather station (latitude: 36.44° S; longitude: 145.27° E; elevation: 114 m), located within the study area.

## 2. Materials and Methods

This study was undertaken in three broad steps. First, the satellite images were processed to generate the relevant measures for a representative date (8 January 2021) for the summer season. Second, a thresholding process was adopted to identify relative differences in surface temperature (Ts−Ta), surface moisture (STR) and vegetation status, which were then used to identify irrigated pixels. As a third step, the classified pixels were aggregated at farm scale so as to match the irrigation water supply information and to measure the classification accuracy.

### 2.1. Satellite Data

A Landsat-8 Satellite Scene (Path 93/Row 85) covered the study area. The image of this scene was taken from Landsat-8 Collection-2 Tier-1, which was calibrated as top-of-atmosphere (TOA) reflectance as follows [35]:

$$\rho\lambda = \frac{\pi \cdot L_\lambda \cdot d^2}{\text{ESUN}_\lambda \cdot \cos\theta_s} \tag{1}$$

where

$\rho\lambda$ = TOA reflectance [-];
$\pi$ = Constant equal to ~3.14159 [-];
$L_\lambda$ = Spectral radiance [W/(m$^2$ sr µm)];
$d$ = Earth-Sun distance [Astronomical units];
$\text{ESUN}_\lambda$ = Mean solar irradiance [W/(m$^2$ µm)];
$\theta_s$ = Solar zenith angle [Degrees].
Pixels with clouds, cloud shadows and cirrus clouds were masked out in the selected image.

### 2.2. Calculation of Surface Temperature (Ts)

At-sensor brightness temperature (Ts) was calculated using the following formula [35]:

$$\text{Ts} = \frac{K2}{nl\left(\frac{K1}{L_\lambda} + 1\right)} \tag{2}$$

where
Ts = Effective at-sensor brightness temperature [K];

$K2$ = Calibration constant 2 [K];
$K1$ = Calibration constant 1 [W/(m$^2$ sr μm)];
$L_\lambda$ = Spectral radiance at the sensor's aperture [W/(m$^2$ sr μm)];
$nl$ = Natural logarithm.
For this study, Band 11 TIR2 (11,500–12,510 nm) was used for $L_\lambda$.

### 2.3. Surface–Air Temperature Difference (Ts−Ta)

Surface temperature has long been used to understand vegetation water status [36]. As plants transpire, water loss reduces leaf temperature due to evaporative cooling. The surface temperature of vegetation often becomes much lower than that of the surrounding air in situations of optimum water supply. Conversely, in water-stressed situations, vegetation transpires less and vegetation surface temperature increases, usually rising above the surrounding air temperature [37]. The difference between the surface and air temperatures (Ts−Ta) is therefore considered a logical variable to assess vegetation water status.

The air temperature data, close to the time of the satellite overpass, were sourced from the Bureau of Meteorology (www.bom.gov.au, accessed on 8 December 2023) and the SILO website (www.longpaddock.qld.gov.au/silo/, accessed on 8 December 2023) for multiple weather stations across and surrounding the study area. The point data of air temperature (Ta) were rasterized, using inverse distance-weighted (IDW) method to match the Ts from satellite.

### 2.4. Calculation of Shortwave Infrared-Transformed Reflectance (STR)

The concept of STR was proposed by Sadeghi et al. [25] in relation to soil moisture. The derivation of STR is based on a simple model of Kubelka–Munk theory [38], which describes radiative transfer with absorption ($k$) and scattering ($s$) in a soil layer, considering a downward and an upward light propagation flux perpendicular to the layer. By analytical solution of the equations for absorption and scattering coefficients, reflectance ($R$) is obtainable as the function of $k$ and $s$ [39]:

$$R = 1 + \frac{k}{s} \sqrt{\left(\frac{k}{s}\right)^2 + 2\frac{k}{s}} \tag{3}$$

The rearrangement of Equation (3) gives 'transformed reflectance' ($r$) denoted by $r = k/s$ [25]:

$$r = \frac{k}{s} = \frac{(1-R)^2}{2R} \tag{4}$$

Sadeghi et al. [25] tested Equation (4) for soil moisture using multiple spectral frequencies in laboratory conditions for a range of soil types, and found that SWIR bands (wavelength between 1100 and 3000 nm) are optimal for soil moisture detection. Therefore, Equation (4) was qualified as 'SWIR-transformed reflectance' (STR) [40]:

$$STR = \frac{(1-SWIR)^2}{2(SWIR)} \tag{5}$$

In the calculation of STR, commonly used spectral bands include Landsat-8 Band 7, Sentinel-2 Band 12 and MODIS Band 7 [40–44]. For this study, *SWIR* in Equation (5) refers to Landsat-8 Band 7 (2200 nm). High STR values indicate high soil moisture and vice versa.

### 2.5. Calculation of Vegetation and Water Indices

Normalized difference vegetation index (NDVI) was calculated as follows [22,45]:

$$NDVI = \frac{NIR - Red}{NIR + Red} \tag{6}$$

where *NIR* denotes Band 5 (865 nm) and *Red* Band 4 (655 nm). NDVI was one of two key variables in land cover classification.

Normalized difference water index (NDWI) was calculated as follows [27]:

$$\text{NDWI} = \frac{Green - NIR}{Green + NIR} \tag{7}$$

where *Green* denotes Band 3 (560 nm). NDWI was used to detect and mask out surface water to ensure the STR estimates were confined to soil moisture.

### 2.6. Thresholding Process for Landcover Classification

An operational classification approach based on iterative thresholds was adopted for this study. This approach, which generates bilevel classes of satellite-derived measures, has been previously used for landcover classification in the irrigation regions [46,47]. Here, we describe the process of both approaches: optical and thermal.

It is widely accepted that an 'irrigated' crop has high vegetation and high soil moisture as compared to non-irrigated crops or other dry landcovers. On this basis, STR (soil moisture indicator) and NDVI (vegetation status) were segmented into two classes each. STR classes referred to 'irrigated' (relatively high soil moisture) and 'non-irrigated' (relatively low soil moisture), and vegetation classes were 'crop' (relatively high NDVI) and 'non-crop' (relatively low NDVI). An iterative thresholding method was used to achieve the binary classification by minimizing within-class variance, $\sigma^2_{Within}$ [48]:

$$\sigma^2_{Within}(T_i) = \omega_0 \sigma^2_0(T_i) + \omega_1 \sigma^2_1(T_i) \tag{8}$$

where $(T_i)$ is the threshold which varies by iteration *i*; $\omega_0$ and $\omega_1$ are the weights of the two classes; and $\sigma^2_0$ and $\sigma^2_1$ are the variance of the two classes.

For applying the thresholding procedure in operation, the relationship of $\sigma^2_{Within}$ with between-class variance ($\sigma^2_{Between}$) and total variance ($\sigma^2_{Total}$) was used as follows [48]:

$$\sigma^2_{Within}(T_i) = \sigma^2_{Total}(T_i) - \sigma^2_{Between}(T_i) \tag{9}$$

Initial NDVI threshold ($\alpha$) was taken as 0.4. All pixels $\geq \alpha$ were considered as 'crop'. Initial soil moisture threshold was the median value of STR ($\varphi$) or Ts−Ta ($\beta$). The iteration interval was set at 0.005 within the limit of $\pm 0.025$ of the initial thresholds. Altogether, 11 iterations each for NDVI and STR/Ts−Ta were performed for the image. Thresholds with minimum $\sigma^2_{Within}$ were used for binary classification.

In the context of the 'optical approach', each pixel was identified with the defined binary classes of NDVI and STR. This resulted in four combinations, as shown in Figure 3A. Pixels in sector (1) denote dry conditions with no or low vegetation. Pixels in sector (2) denote wet conditions with low or no vegetation. Pixels in sector (3) indicate some vegetation, which may be crop or pasture but without irrigation. Sector (4) denotes vegetation with wet condition indicating irrigated crop or pasture.

In the context of the 'thermal approach', the thresholding process for both NDVI and Ts−Ta was the same as described earlier. As shown in Figure 3B, the four sectors define the vegetation status and wet/dry conditions. Here, the pixels in Sector (4) indicate irrigated crop or pasture.

Validation was carried out on the pixels aggregated to the extent of irrigation water delivery licences. Water delivery for each license has been linked to spatial unit consisting of one or more paddocks. These units hereinafter are referred to as farms. The mapped results of irrigated landcover were compared with the actual irrigation deliveries to farms. Information on irrigation water supplies at farm level was sourced from the Victorian Water Register (VWR), a state-wide irrigation water database (https://waterregister.vic.gov.au/, accessed on 8 December 2023). Figure 4 shows the location of farms which have been included in the validation process. In this figure, the locations of farms have been shown by farm centroids instead of farm boundaries due to privacy reasons.

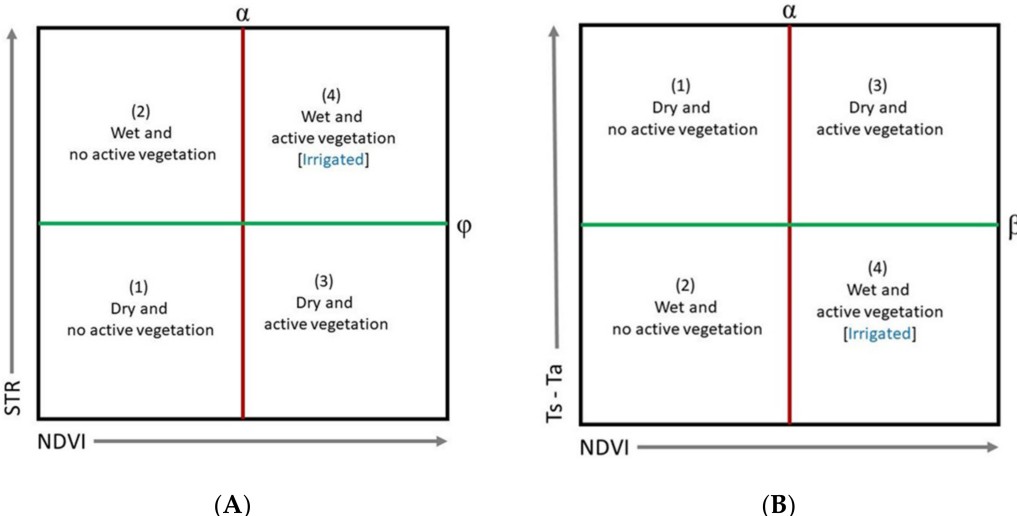

(**A**)                                                          (**B**)

**Figure 3.** (**A**) Concept of thresholding classification (optical approach). α and φ are the thresholds for NDVI and STR, respectively, determined by an iterative thresholding method by minimizing within-class variance. Irrigated pixels are those with NDVI ≥ α and STR ≥ φ located within Sector 4. (**B**) Concept of thresholding classification (thermal approach). α and β are the thresholds for NDVI and Ts−Ta, respectively, determined by an iterative thresholding method by minimizing within-class variance. Irrigated pixels are those with NDVI ≥ α and Ts−Ta ≤ β in Sector 4.

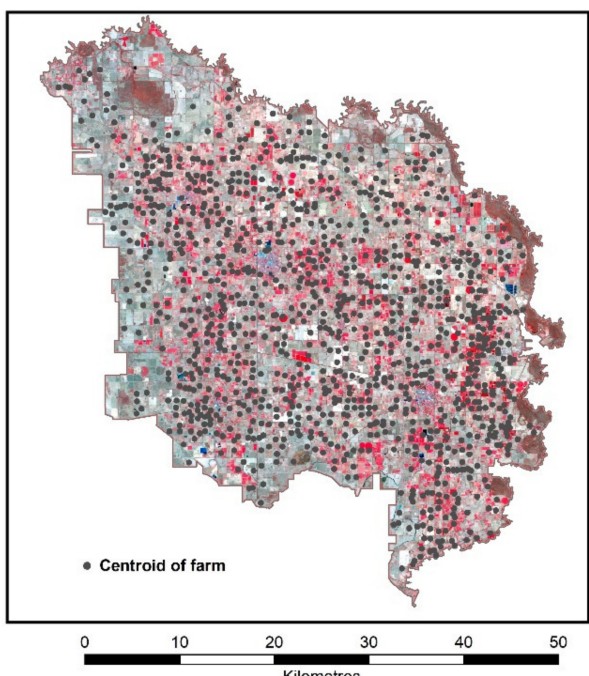

**Figure 4.** Centroids showing the location of the farms in CGID used in the validation process. The background is the Landsat-8 images with bands 5/4/3 (displayed as a false colour composite image).

## 3. Results

### 3.1. Distribution of Input Measures

The classification process takes two variables as inputs in both approaches—'optical' and 'thermal'. NDVI is one variable, which is a common measure used in both approaches, as shown in Figure 5A. Although the NDVI ranges between 0 and ≥0.8 in CGID (Table 2), the notable areas are those with NDVI ≥ 0.4, shown as shades of green (Figure 5A). These areas are predominantly under perennial horticulture, perennial pastures or summer crops.

The NDVI distribution appears patchy and irregular, mainly due to the widely varying sizes of agricultural properties ranging between under 2 ha and over 1000 ha.

**Figure 5.** Maps of individual input measures in CGID. (**A**) NDVI, (**B**) STR, (**C**) NWDI, (**D**) Ta, (**E**) Ts and (**F**) Ts−Ta.

**Table 2.** Statistics of individual input measures used in classification.

| Statistics | NDVI | STR * | NDWI | Ta | Ts | Ts−Ta |
|---|---|---|---|---|---|---|
| Minimum | 0.000 | 0.000 | −0.776 | −14.938 | 0.000 | 0.000 |
| Mean | 0.286 | 1.800 | −0.379 | 22.963 | 38.304 | 15.125 |
| Median | 0.239 | 1.356 | −0.355 | 22.923 | 38.754 | 15.562 |
| Maximum | 0.875 | 36.696 | 0.818 | 36.813 | 47.227 | 24.081 |
| SD | 0.126 | 1.457 | 0.101 | 0.917 | 2.7125 | 2.455 |

* Excluding pixels with 'surface water' (NDVI < 0.1 and NDWI > 0.1).

The values of STR in CGID range between 0 and ≥36 (Table 2). However, the notable areas are those with STR values of 2 and over, indicating the presence of substantial surface moisture (Figure 5B). Here, the distinction must be made between potential irrigation areas, which are exclusively agricultural properties, and the riverine zone of native vegetation, where the source of surface moisture is not irrigation.

Figure 5C shows the distribution of NDWI in CGID. All NDWI values above zero indicate the presence of surface water (Table 2). In addition to a few large water bodies like Waranga Basin in the south-central part, farm dams of varying sizes are shown across the district. The areas with surface water were excluded while evaluating STR values.

The 'Ts−Ta' input variable for the thermal approach of classification is largely driven by the satellite-derived surface temperature (Ts), moderated by air temperature (Ta). The spatial variation of Ta in CGID was within 3°–4 °C (Table 2). It ranged between under 21 °C and approximately 24 °C, increasing from the south-east to the north-west direction (Figure 5D). On the other hand, the Ts ranged between 26° and over 40 °C (Table 2). Relatively cooler surfaces were with <34 °C, roughly spread in patches but largely contiguous in the riverine areas in the east (Figure 5E).

The distribution of Ts−Ta in CGID ranged between 0° and 24 °C (Table 2, Figure 5F). Here, the relatively cooler areas appear more pronounced as compared to those in the Ts.

A visual comparison of the Ts−Ta map (Figure 5F) with that of the STR (Figure 5B) shows marked similarities. The locations of high surface moisture (i.e., high STR and low Ts−Ta) appear to be the same. Similarly, the dry surface areas of the two maps correspond well.

### 3.2. Thresholds for Pixel Classification

Figure 6A,B show the identification of pixels as per the optical and thermal approaches. Using Equation 9, the iteration process provided the threshold for the NDVI at 0.425, for the STR at 1.425 and for the Ts−Ta at 14.85 °C. In each case, the pixels were divided into four classes. Pixels within Sector Four (refer to Figure 3A,B) were identified as 'irrigated' crop or pasture.

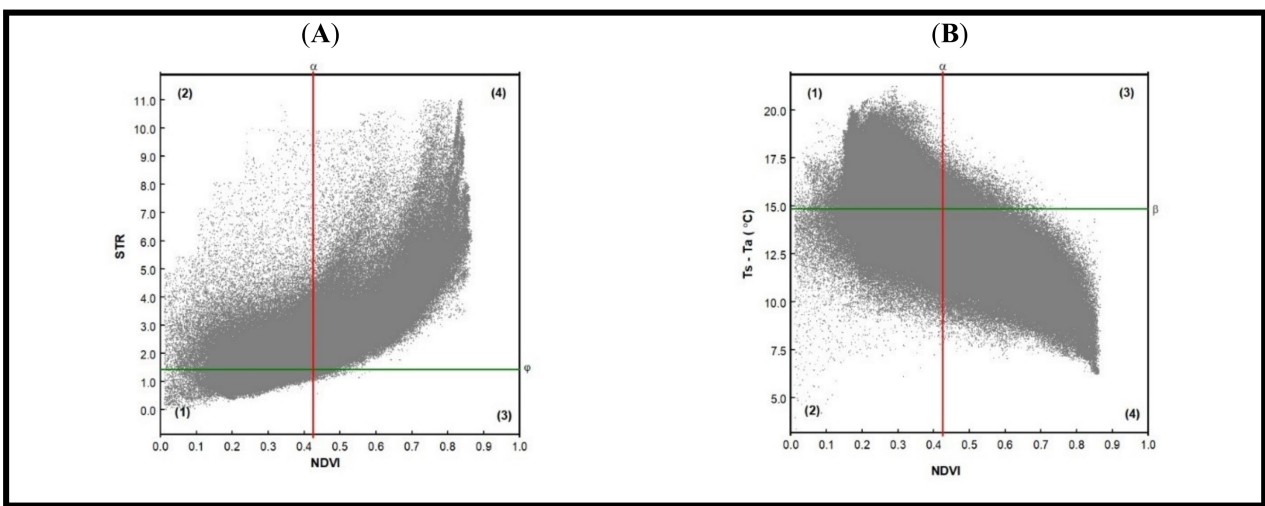

**Figure 6.** Pixel identification based on thresholds. (**A**) Optical approach and (**B**) thermal approach of individual input measures in CGID. (Refer to Figure 3A,B for the descriptions of the four sectors, labelled here as 1–4.)

### 3.3. Irrigated Landcover Classification

Figure 7 shows the results of the irrigated landcover classification for the optical approach. Figure 7A shows the four classes as identified in Figure 6A. Figure 7B is the map of the irrigated landcover class only. Figure 8 shows the results of the irrigated landcover classification based on the thermal approach, where Figure 8A shows the four classes as identified in Figure 6B. Figure 8B is the map of the irrigated landcover class only.

A visual comparison of the two maps of irrigated landcover (Figures 7B and 8B) shows minor differences. The map of the optical approach (Figure 7B) shows slightly more pixels (13.86% of the total area) as irrigated in comparison to the map of the thermal approach (12.43% area) (Figure 8B, Table 3).

**Table 3.** Area of landcover classes in CGID.

| Classes | Thermal Approach | | Optical Approach | |
|---|---|---|---|---|
| | Hectares | % Area | Hectares | % Area |
| Irrigated: Wet Vegetation | 23,651 | 12.43 | 26,370 | 13.86 |
| Dry Vegetation | 2835 | 1.49 | 116 | 0.06 |
| Wet No Vegetation | 29,118 | 15.30 | 60,512 | 31.80 |
| Dry No Vegetation | 134,684 | 70.78 | 103,289 | 54.28 |

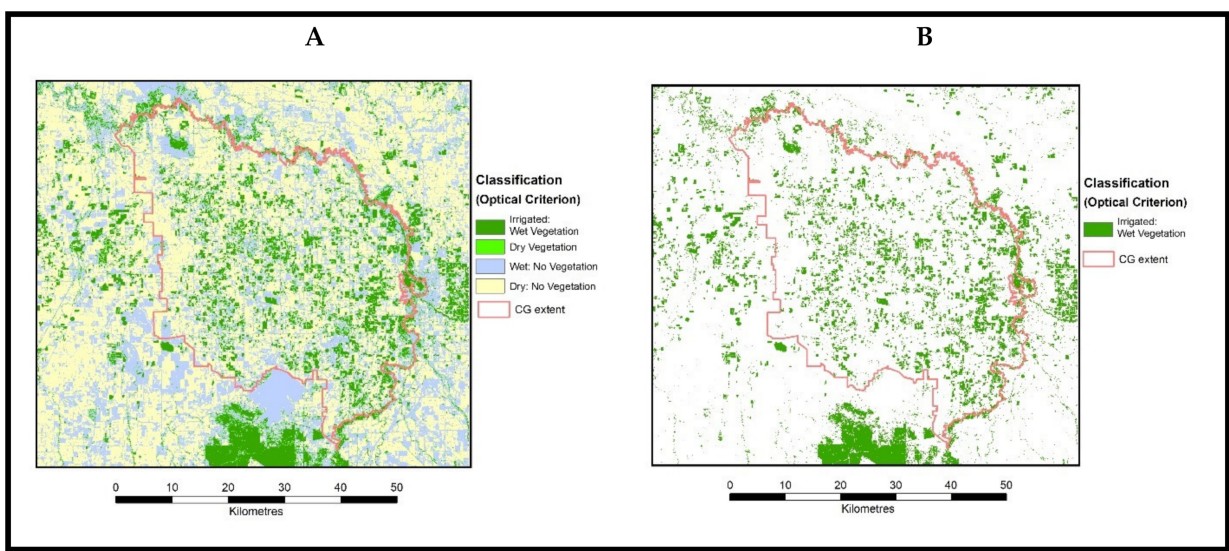

**Figure 7.** Maps of irrigated landcover classes (optical approach). (**A**) Pixel classes, (**B**) irrigated landcover.

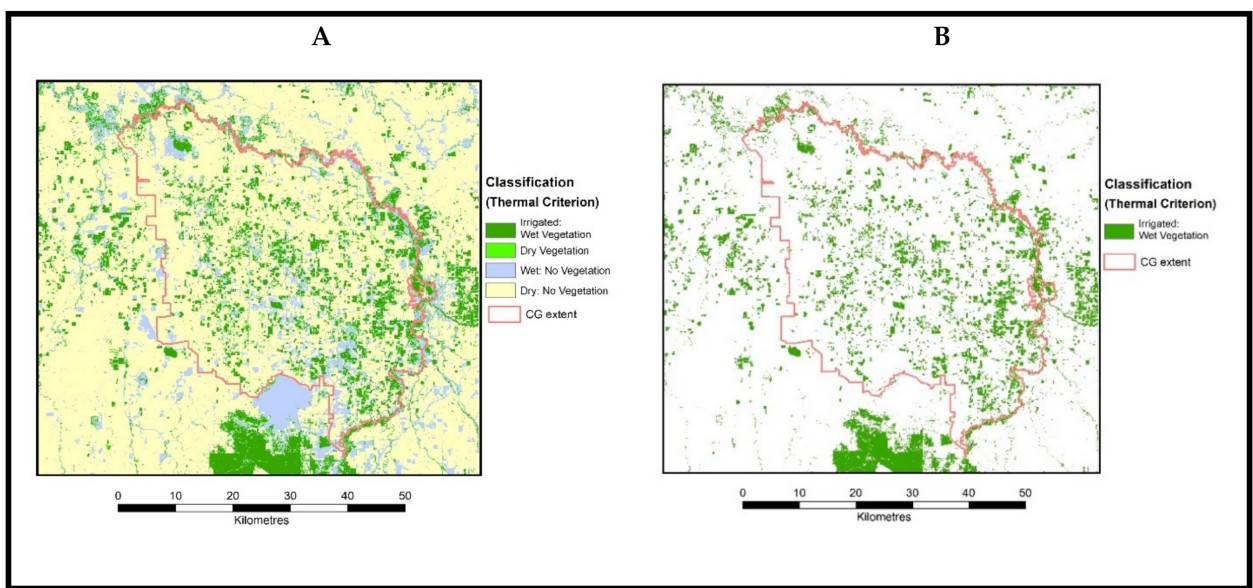

**Figure 8.** Maps of irrigated landcover classes (thermal approach). (**A**) Pixel classes, (**B**) irrigated landcover.

*3.4. Validation of Irrigated Landcover*

As stated earlier, validation of the irrigated land cover classification was carried out at the farm level. Farms ≥ 10 ha in size were selected for validation. A farm was considered as 'irrigated' ('Reference') based on a substantive amount of water delivery (≥0.1 ML per ha) to that farm during the timeframe of the summer season (December to February). No or a negligible amount of water delivery (<0.1 ML per ha) was taken as the absence of irrigation on a farm. These reference farms were compared with the classification results for accuracy.

As per the classification, a farm was considered 'irrigated' provided a substantial proportion of pixels (≥25%) within that farm belonged to Sector 4 (wet and active vegetation; refer to Figures 3 and 6). This criterion was uniformly applied to both the optical and thermal approaches.

Tables 4 and 5 provide the classification accuracy for the two approaches. The calculation of the confusion matrix resulting in the producer's and user's accuracies followed

the standard procedure [49,50]. Although the overall accuracy was high in both cases, the optical approach (97.6%) performed slightly better than the thermal approach (93.9%). The user's accuracy for irrigated landcover was noticeably higher in the case of the optical approach (98.3%) as compared to that of the thermal approach (88%). However, there was very little difference between the two approaches as per the producer's accuracy for irrigated landcover (96.2% for the optical and 97.1 for the thermal approach).

**Table 4.** Evaluation of irrigated landcover classification (Optical approach).

| (a) Occurrence (Optical approach) | | | |
|---|---|---|---|
| Classification | Reference * | | Total |
| | Irrigated | Non-Irrigated | |
| Irrigated | 451 | 8 | 459 |
| Non-Irrigated | 18 | 624 | 642 |
| Total | 469 | 632 | 1101 |
| * Classification according to water delivery records | | | |
| (b) Accuracy (Optical approach) | | | |
| | Producer's Accuracy % | | User's Accuracy % |
| Irrigated | 96.2 | | 98.3 |
| Non-Irrigated | 98.7 | | 97.2 |
| Overall accuracy = 97.6% | | | |

**Table 5.** Evaluation of irrigated landcover classification (Thermal approach).

| (a) Occurrence (Thermal approach) | | | |
|---|---|---|---|
| Classification | Reference * | | Total |
| | Irrigated | Non-Irrigated | |
| Irrigated | 404 | 55 | 459 |
| Non-Irrigated | 12 | 630 | 642 |
| Total | 416 | 685 | 1101 |
| * Classification according to water delivery records | | | |
| (b) Accuracy (Thermal approach) | | | |
| | Producer's Accuracy % | | User's Accuracy % |
| Irrigated | 97.1 | | 88.0 |
| Non-Irrigated | 92.0 | | 98.12 |
| Overall accuracy = 93.9% | | | |

## 4. Discussion

The feasibility of using the STR along with the NDVI (optical approach) to map irrigated landcover has been confirmed by a high-level overall accuracy assessment (97.6%). These results of irrigated landcover by an optical approach were found to be comparable with those of a parallel process that used the Ts−Ta along with the NDVI (thermal approach). However, there was a noticeable difference in the user's accuracy for the 'irrigated' class between the two approaches (88% thermal to 98.3% optical). This could have occurred due to the difference in the spatial resolution of a variable for surface moisture. The STR is based on Band 7 of Landsat 8 which has a 30 m resolution, whereas Band 11, which is used in the Ts−Ta, is at 100 m resolution, though it is resampled to 30 m [51]. For the same reason, the 'wet no vegetation' areas were estimated higher by the optical approach than by the thermal approach (Table 3).

The STR is expected to work in all ground conditions, since it has been tested by Sadeghi et al. [25] for a range of soil types, and was used in this study successfully.

The overriding objective of our paper was to assess the feasibility of using the STR as a soil moisture measure, in place of the TIR-based measure. We attempted to do that by using a simple but effective method of classification which has been used previously. Our iterative thresholding procedure was based on minimizing the within-class variance for bilevel segmentation. This step is like Otsu's method [48]. However, our approach did not include an automatic optimization process unlike Otsu's method. Otsu's method, with an automatic optimization process, performs well to find a single threshold only when the histogram has a distinct bimodal distribution. In other situations, it does not deliver an optimal result. The variables for our study area were not bimodal. Therefore, we introduced certain constraints to the method, i.e., an initial threshold and iteration limits, as described earlier in Section 2.6.

The initial threshold for the NDVI (0.4) was on the assumption that, in an agriculture enterprise, vegetation with NDVI < 0.4 is unlikely to be a managed crop or pasture. In the case of the STR and the $Ts-Ta$, the initial threshold taken was the median pixel value of the variable, which was based on the experience from the previous studies in the region [46,47,52]. These initial thresholds and the related assumptions should be field tested further to ascertain the strengths and weaknesses of this approach in different landscapes and at different times.

This study used a simple threshold method for classification. However, for future studies, it is desirable to employ other approaches such as machine learning over larger areas.

The classification method used in this study assumed that the main source of surface moisture was irrigation and that rainfall did not have any significant impact. As stated earlier, the summers in the study area are generally dry. The long-term average of rainfall during the three months of summer is 101.8 mm (Figure 2). The study period (December 2020 to February 2021) was, however, drier, with a total of 80.6 mm. In cases of high rainfall situations, there will be a requirement for season-wide vegetation as well as STR analysis of the known irrigated and dryland farms vis-à-vis rainfall records to make the desired adjustments to the thresholding method.

This study used a single image as a representative of the summer season. This is in accordance with a previous study in the irrigation regions of Victoria, Australia, which found the use of a single 'mid-season' image adequate to assess maximum crop cover because of the strong temporal stability in NDVI response [53]. However, for further studies, there is scope to test temporal aggregates of image-based variables from multiple sources including Landsat-8/9 and Sentinel-2.

This study is a formal affirmation of using the STR as a substitute for Ts-based measures for irrigated landcover classification. The basis of this affirmation is the formal comparison of the optical approach with the thermal approach implemented in a key irrigation district. This opens the opportunity of utilizing the widely available optical sensors for irrigated landcover investigations.

## 5. Conclusions

The satellite-derived surface moisture (STR, Ts) and vegetation status (NDVI) are appropriate measures to distinguish between irrigated and non-irrigated pixels within farmlands. The use of the STR, which is based on Band 7 of Landsat-8 (the optical spectral range), with the NDVI, demonstrated successful mapping of irrigated landcover for the summer season of 2020–2021 in CGID. The results of this optical approach of classification were found to be comparable with those of the thermal approach, where $Ts-Ta$ is used with the NDVI for classification. These results have broader implications in terms of irrigated landcover assessment, as the use of satellite imagery in these applications may not necessarily be limited to sensors with either microwave or thermal bands. The impact of this is a greater freedom, both spatially and temporally, to develop this information.

**Author Contributions:** Conceptualization, M.A. and A.M.; methodology, M.A.; validation, A.M., M.A. and K.S.; formal analysis, M.A.; resources, A.M.; writing—original draft preparation, M.A.; writing—review and editing, K.S. and M.A.; project administration, A.M. and M.A. All authors have read and agreed to the published version of the manuscript.

**Funding:** This research received no external funding.

**Data Availability Statement:** The data sets are not available due to privacy reasons for individual farm holders.

**Acknowledgments:** The authors are thankful to the leadership team of Agriculture Victoria Research for their support of the work.

**Conflicts of Interest:** The authors declare no conflicts of interest.

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
