# Peer review of "Feasibility of Using SWIR-Transformed Reflectance (STR) in Place of Surface Temperature (Ts) for the Mapping of Irrigated Landcover"

_land, doi:10.3390/land13050633_

Round 1

Reviewer 1 Report

Comments and Suggestions for Authors

The nice aspect of this paper is the comparison among a straightforward classification algorithm by thresholding for SWIR Transformed Reflectance vs Surface Temperature and their comparison against ground truths. While a more refined ML approach could even be considered to improve results, this work demonstrates the capabilities of a pure physics-based methodology, I would say. I would be curious to see results on a larger scale and increased resolutions.

Author Response

Authors are thankful to the reviewer for the kind and encouraging comments. Text has been added to the discussion to make the point that it is desirable to employ other approaches for landcover classification such as machine learning for future studies over larger areas.

Reviewer 2 Report

Comments and Suggestions for Authors

The research presented in this manuscript focuses on comparing two methods of recognizing irrigated areas using satellite data.

The topic is current and of interest to the scientific community in mainly allocative terms. The methodologies presented for the recognition of irrigation status are not new, but the experimentation comparing the method based on optical data and the method based on thermal data presents very useful results.

The following are some suggestions for improving the study:

-It might be helpful to better understand how to test the proposed method in other areas to have a brief description of the geological and pedological context of the soils involved in the study;

- delete in line 114 a "The"

- correct the wavelength range of the SWIR bands (row 173)

- add a table/graph comparing the results, referring to Figures 5, 7 and 8 (currently only a visual analysis is proposed, which is not very effective).

Author Response

Authors are grateful to the reviewer for the kind comments and constructive suggestions.

Text has been added to Section 1.1 for the study area to describe its geological and pedological background.

The approach using STR is expected to work in all ground conditions since STR has been tested for a range of soil types by the authors (Sadeghi et al.) who introduced the measure. This point has been added to the Discussion section.

The extra word ‘The’ has been deleted from line 173 as advised.

On line 173, the wavelength of SWIR range has been corrected.

As suggested, the following two new tables have been added to the Result section and the text has been changed as appropriate:

Table 2. Statistics of individual input measures used in classification.

Table 3. Area of landcover classes in CGID.

Reviewer 3 Report

Comments and Suggestions for Authors

The manuscript is well-constructed and the approaches developed was clearly stated. However, please consider the following comments for improving your work:

1) Please make sure there is no grammar errors and improve the writing.

2) Please enhance the introduction carefully. The current version has flaws that needs to be addressed. 

3) For your iterative threshold method, have you ever consider the Otsu's method that can automate the process of finding the optimal threshold.

Again, improve the writing (especially rewrite introduction) and consider to compare your method to Otsu's method.

Comments on the Quality of English Language

Need to work hard to improve the writing.

Author Response

Authors are thankful to the reviewers for the constructive comments.

Authors have read the manuscript carefully and made the necessary changes to the text for clarity and grammar.

Minor errors in the text of the Introduction section have been corrected.

Our iterative thresholding approach to minimize within-class variance for bilevel segmentation is like Otsu’s method. However, our approach does not include automatic optimization process unlike Otsu’s method. Text has been added to clarify this point.

Round 2

Reviewer 3 Report

Comments and Suggestions for Authors

In the first round of review, I suggested the authors comparing their threshold selection method to Otsu's method. In their revised Discussion section, they responded, stating, "This step resembles Otsu's method [48]. However, our approach doesn't involve an automatic optimization process like Otsu's method."

If the key part of your approach shares similarities with Otsu's method but less efficient, why didn't you used Otsu's method? Please give more arguments. 

Also, in the same section, you suggested that "However, for future studies, it is desirable to employ other approaches such as machine learning over larger areas." Why did you believe machine learning approaches can outperform your work? It seems unnecessary to bring up machine learning merely because it's a popular topic now.

Author Response

Authors are thankful to the reviewer for the comments.

Please note authors’ response to the comments below.

Otsu's method, with automatic optimization process, performs well to find a single threshold only when the histogram has a distinct bimodal distribution. In other situations, it does not deliver optimal result. Therefore, users have introduced variations to Otsu’s methods as per situations. In the irrigation region in question (this study area), for example, NDVI distribution is not bimodal. This led us to introduce certain constraints (initial threshold and iteration limits) to the method as described in the methodology.

The overriding objective of our paper has been to assess the feasibility of using STR as soil moisture measure, in place of TIR-based measure. We attempted to do that by using a simple but effective method of classification which has been previously used. Our objective in this paper was not to explore the best methods of classification. However, in our view, it is desirable for future to use other methods that can incorporate STR as a variable, just to ascertain the strength and weaknesses of ‘optical approach’ that we have introduced in this paper.  We mentioned ‘machine learning’ just an example.